# Carbonyl sulfide (COS) emissions in two agroecosystems in central France

**Sauveur Belviso**[1]*, **Camille Abadie**[1], **David Montagne**[2], **Dalila Hadjar**[2], **Didier Tropée**[3], **Laurence Vialettes**[1], **Victor Kazan**[1], **Marc Delmotte**[1], **Fabienne Maignan**[1], **Marine Remaud**[1], **Michel Ramonet**[1], **Morgan Lopez**[1], **Camille Yver-Kwok**[1], **Philippe Ciais**[1]

**1** Laboratoire des Sciences du Climat et de l'Environnement, Université Paris-Saclay, CEA-CNRS-UVSQ, UMR8212, Gif-sur-Yvette, France, **2** UMR ECOSYS, Université Paris-Saclay, INRAE, AgroParisTech, Thiverval-Grignon, France, **3** UMR GQE, Université Paris-Saclay, INRAE-CNRS, AgroParisTech, Gif-sur-Yvette, France

\* sauveur.belviso@lsce.ipsl.fr

**Data Availability Statement:** All relevant data are within the manuscript and the publicly available external repository available here: https://mycore.core-cloud.net/index.php/s/wUsUpMYrW9FUniz

## Abstract

Carbonyl sulfide (COS) fluxes simulated by vegetation and soil component models, both implemented in the ORCHIDEE land surface model, were evaluated against field observations at two agroecosystems in central France. The dynamics of a biogenic process not yet accounted for by this model, i.e., COS emissions from croplands, was examined in the context of three independent and complementary approaches. First, during the growing seasons of 2019 and 2020, monthly variations in the nighttime ratio of vertical mole fraction gradients of COS and carbon dioxide measured between 5 and 180 m height ($Grad_{COS}$/$Grad_{CO2}$), a proxy of the ratio of their respective nocturnal net fluxes, were monitored at a rural tall tower site near Orléans (i.e., a "profile vs. model" approach). Second, field observations of COS nocturnal fluxes, obtained by the Radon Tracer Method (RTM) at a sub-urban site near Paris, were used for that same purpose (i.e., a "RTM vs. model" approach of unaccounted biogenic emissions). This site has observations going back to 2014. Third, during the growing seasons of 2019, 2020 and 2021, horizontal mole fraction gradients of COS were calculated from downwind-upwind surveys of wheat and rapeseed crops as a proxy of their respective exchange rates at the plot scale (i.e., a "crop based" comparative approach). The "profile vs. model" approach suggests that the nocturnal net COS uptake gradually weakens during the peak growing season and recovers from August on. The "RTM vs. model" approach suggests that there exists a biogenic source of COS, the intensity of which culminates in late June early July. Our "crop based" comparative approach demonstrates that rapeseed crops shift from COS uptake to emission in early summer during the late stages of growth (ripening and senescence) while wheat crops uptake capacities lower markedly. Hence, rapeseed appears to be a much larger source of COS than wheat at the plot scale. Nevertheless, compared to current estimates of the largest COS sources (i.e., marine and anthropogenic emissions), agricultural emissions during the late stages of growth are of secondary importance.

(https://doi.org/10.14768/6800b065-dcec-4006-ada5-b5f62a4bb832).

**Funding:** The author(s) received no specific funding for this work.

**Competing interests:** The authors have declared that no competing interests exis.

## Introduction

The uptake of atmospheric carbonyl sulfide (COS) through stomata in the plant leaves and subsequent irreversible hydrolysis by enzymatic reaction with carbonic anhydrase ($F_{COSveg.}$) is the largest sink in the global budget of this compound, yet estimates differ by as much as a factor of 5.6 as discussed in a recent review [1]. However, if estimates of $F_{COSveg.}$ based on net primary production (NPP) scaling are excluded, since it is now known that $F_{COSveg.}$ has to be scaled rather to gross primary production (GPP) than NPP, the rest of the estimates based on GPP scaling or mechanistic models vary by only a factor of two [1]. A larger uncertainty is attached to the annual amount of COS exchanged by soil, by as much as a factor of 14 as shown in Table 3 of [2]. However, most recent simulations using mechanistic vegetation and soil COS models of [3–6] implemented in the ORCHIDEE land surface model show that the relative contribution of soils in the global budget is secondary, taking up 19 times less COS from the atmosphere than vegetation on an annual basis (i.e., -30 GgS yr$^{-1}$ vs. -576 GgS yr$^{-1}$, respectively) [2]. This is because models consider (1) that the uptake capacity of oxic soils can be partly countered by a production mechanism, the seasonality of which is mainly driven by temperature, and (2) that anoxic soils behave as sources of COS [5, 6]. Although laboratory and field observations have shown in rare cases that vascular plants could play also a role in COS production [7–9], the processes by which plants emit this gas have yet to be considered in global modeling studies. COS production has also been reported for mosses [10] and lichens [11]. Any method suitable for assessing plant emissions at different spatial scales would help to resolve uncertainties in the global COS budget and to correct the imbalance between bottom-up estimates of total sources and sinks, the latter generally exceeding the former by hundreds of GgS yr$^{-1}$ before correction [12, 13].

Our goal is to examine the capacity of agricultural ecosystems and of some specific crops in producing COS, which in mid-latitude regions exhibit high productivity and cover very large areas. Moreover, shifts from sink to source have already been reported in such ecosystems [8, 14]. For that purpose, we used field observations of (1) vertical and horizontal concentration gradients of COS and (2) COS fluxes obtained by the Radon Tracer Method (RTM). From the comparison of observed and modelled COS fluxes, we propose an empirical function suitable for inventorying crop emissions at different spatial scales.

## Materials and methods

No specific permissions were required for these locations/activities. I confirm that the field studies did not involve endangered or protected species.

### Experimental sites / applied flux quantification methods

Atmospheric boundary layer mixing ratios were monitored at two sites, Gif-sur-Yvette (GIF) and Trainou (TRN tall tower), both in central France (Fig 1).

### TRN site: Nocturnal vertical gradients

The Trainou 180-m tall tower atmospheric observatory is located about 80 km south of GIF (Fig 1). A description of the TRN site is available online at https://icos-atc.lsce.ipsl.fr/panelboard/TRN. At the TRN tall tower, atmospheric COS was measured with the Aerodyne Research quantum cascade laser (mini-QCL) formerly deployed at the SAC station [14]. The sampling lines, made of Synflex tubing, collected air at 4 heights (5 m, 50 m, 100 m and 180 m) sequentially, the total sequence lasting for 80 minutes. Because the TRN station is operated in the framework of ICOS European Infrastructure [15], the mini-QCL was synchronized with

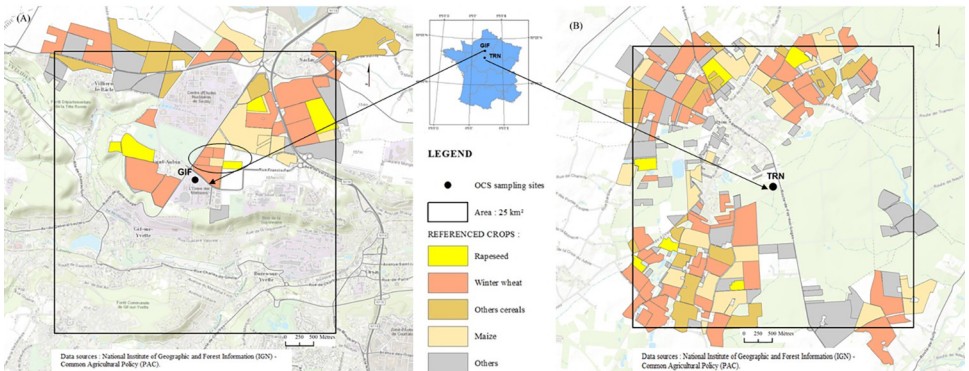

**Fig 1. Maps showing the location of the GIF/TRN sampling sites and Common Agricultural Policy (CAP) declared crops (year 2019).** The black squares delineate an area of 25 km$^2$ in the immediate proximity of both stations. The ellipse on the left map delineates the agricultural fields where the uptake/emission of COS has been surveyed for 3 consecutive years. The winter rapeseed (WR) plot area was 4.5 ha in 2019. In 2020 and 2021, WR was grown at the maize (MI) plot. The winter wheat (WW) plots sampled in 2020 and 2021 are those located NE and W of the MI field, respectively. Both have the same size, 3.5 ha each. This figure is similar but not identical to the original image and is therefore for illustrative purposes only.

an ICOS analyzer measuring $CO_2$, $CH_4$, CO and $H_2O$ (Picarro Model G2401). A proxy of the ratio of nocturnal ecosystem fluxes ($F_{COS}/F_{CO2}$) was calculated from nighttime vertical gradients of COS and $CO_2$ ($Grad_{COS}/Grad_{CO2}$ from data collected at 5 m and 180 m height), assuming similarity of COS fluxes and gradients to $CO_2$ fluxes and gradients. Data used to calculate the linear regression slope between $Grad_{COS}$ and $Grad_{CO2}$, were selected according to time of day as follows: [21; 4] (UTC) in spring and summer since we only aimed at examining the capacity of local agricultural ecosystems in exchanging COS with the atmosphere. The linear regression slope between $Grad_{COS}$ and $Grad_{CO2}$ is forced through zero because vertical gradients tend to zero during windy nights. The relative vertical gradients of COS measured at TRN between 5 and 180 m can be used to assess local exchange rates provided that the highest inlet height remains within the nocturnal boundary layer (NBL) throughout the night. This necessary condition was achieved during the months of May through August of 2019 and 2020 as shown later from an analysis of the diurnal variations in monthly mean $CO_2$ profiles. Emission and uptake rates of COS from nearby crops were not investigated from horizontal mole fraction gradients there.

## GIF site: RTM method and horizontal concentration gradients

The GIF monitoring station is located at LSCE, l'Orme des Merisiers, about 2 km south of the Saclay station (SAC) part of the Integrated Carbon Observation System (ICOS) network (https://icos-atc.lsce.ipsl.fr/panelboard/SAC), and next to the village of Saint Aubin. For more information about land cover classification, see Fig 1 of [14] and the EU Common Agricultural Policy (CAP) maps shown in Fig 1.

At GIF, atmospheric COS was measured at 7 m, on line every hour using an automated sampling system (Entech P7100) coupled to a gas chromatograph (GC, Varian 3800) [14, 16]. The GIF time series currently covers seven years (from August 2014 to September 2021). COS fluxes were computed with RTM as by [14], an approach assuming comparably homogeneous spatial distributions of the soil $^{222}$Rn source and the ecosystem COS sink/source. COS fluxes are computed as the product of COS/$^{222}$Rn slopes during nighttime inversion and $^{222}$Rn local exhalation rates. The latter (i.e., 52 Bq m$^{-2}$ h$^{-1}$ with a seasonal cycle amplitude of +25% in summer and -25% in winter driven by soil moisture) was taken from [17]. The RTM allowed us to

make 400 determinations of nocturnal fluxes of COS throughout the 2014–2021 period as detailed later.

The small ellipse on the left panel of Fig 1 delineates the fields from which COS fluxes were assessed indirectly from downwind minus upwind horizontal concentration gradients along the wind direction (WD). First, wind speed (WS) and WD data from a weather station located on the LSCE roof top (Vaisala model WXT530) was used to approximate local meteorology. In the field, we used the facilities of a local meteorological station belonging to the Institut National de la Recherche Agronomique (INRA), equipped with a wind vane providing a continuous visualization at the local scale of the WD. WS data from the INRA weather station was downloaded later. Horizontal COS concentration gradients were determined discontinuously in spring-summer of 2019, 2020 and 2021. They were documented between 9:00 and 11:00 AM (local time), during non-rainy days, from flask-air samples collected in pairs each upwind and downwind ($2 < WS < 14$ km h$^{-1}$, mean = 6 km h$^{-1}$, SD = 3.6 km h$^{-1}$) of selected crops fields surrounded by the ellipse in Fig 1A. Although the total duration of the crop surveys was 2 hr, each horizontal gradient was documented within 40 minutes. We ensured that the concentration gradient was not affected by the residual nocturnal boundary layer by comparing samples collected on an hourly basis at 7 m (GIF time series) with upwind flask samples collected at canopy height. The sampling device has been described by [18]. Once filled with sample air, flasks were transferred within hours to the laboratory nearby and analyzed for COS with the Entech/Varian instruments described below. Precision for GC measurements of flask-air samples is reported as box plots in Fig 2.

During May to July of 2021, 29 pairs of flasks were collected and each flask was analyzed twice consecutively for COS. The average difference between duplicated analyses was $9.3 \pm 7.2$ ppt (median = 7.5 ppt, interquartile range (IQR) = 4.5–12.5 ppt, n = 58). The average difference between flasks of the same pair, assessed as the median of flask 1 minus that of flask 2, was $5.5 \pm 4.5$ ppt (median = 4.9 ppt, IQR = 2.2–8.0 ppt, n = 29). In 90% of the analyses, the difference between flasks of the same pair was less than 14 ppt.

Our study of uptake/emission from the rapeseed and wheat fields covers a measurement period from March 30[th] to July 29[th], 2021, encompassing most important growth stages, i.e., stem elongation/extension, flowering, ripening and senescence. Winter rapeseed (*Brassica napus*, Blackbuzz variety) was sown on September 5[th], 2020 and harvested August 6[th], 2021. A mixture of four varieties of winter wheat (*Triticum aestivum*; Renan, Gwenn, LG Absalon and Chevignon varieties) was sown on October 31[st], 2020 and harvested August 14[th], 2021. Agricultural treatments, including herbicides, insecticides and nitrogen fertilizers, were applied to each crop in autumn and winter. The wheat field was fertilized with sulfur in early March 2021. Wheat and rapeseed yields were 6.8 and 3.7 t ha$^{-1}$ in 2021, respectively. COS exchanges could not be investigated in 2020 as in 2021 because of the French COVID-19 lockdown. All the cultivated plots sampled in our study area, which are less than 500 meters apart (Fig 1A), show very similar soils and the rotation of cultivars is a general feature of agricultural practices in the GIF area.

## Simulations of ecosystem fluxes at the GIF and TRN sites

The ORCHIDEE Land Surface Model is developed at the Institut Pierre Simon Laplace (IPSL). It computes the carbon, water and energy balances over land surfaces [19]. Biome types are grouped into 15 Plant Functional Types (PFTs), including bare soil. At the GIF and TRN sites, the vegetation distribution was prescribed based on the respective land cover map and the soil texture is defined as silt loam in texture classification of the United States Department of Agriculture (USDA) [14].

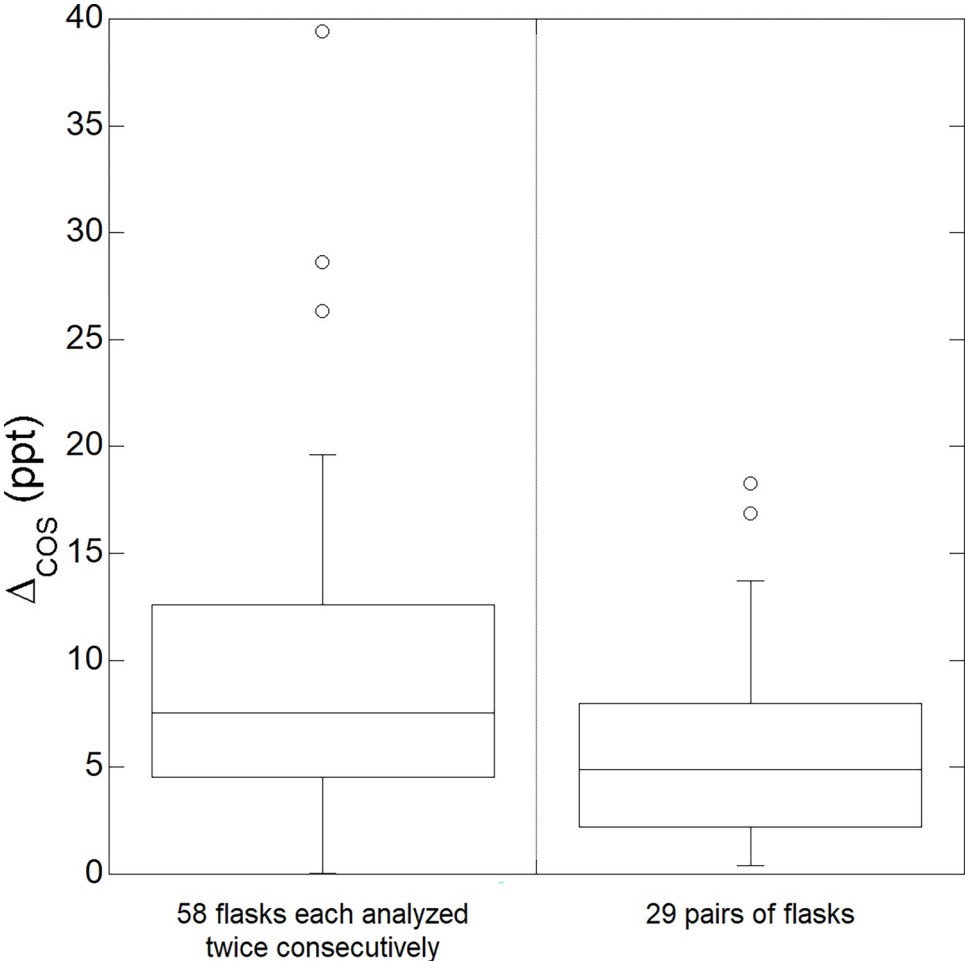

**Fig 2. Analysis precision for COS measurements from flask-air samples collected upwind and downwind of selected crops.** Left box plot: difference in COS (ppt dry air) between two consecutive analyses of each flask content (n = 58). Right box plot: difference in COS (ppt dry air) between flasks of the same pair (n = 29). Shown are 10th, 25th, median, 75th and 90th percentile. Circles correspond to outliers. The study period extends from late March to late July 2021.

ORCHIDEE was forced by 0.25˚x0.25˚ hourly reanalysis fields from the fifth generation of meteorological analysis of the European Centre for Medium-Range Weather Forecast (ECMWF ERA5, [20]). Near-surface COS concentrations were prescribed using monthly averages of tower atmospheric concentration measurements at these sites for the available years, or using simulated monthly average concentrations by the Laboratoire de Météorologie Dynamique atmospheric transport model (LMDz, [13]). Near-surface $CO_2$ concentrations are estimated using global annual-mean values provided by the TRENDY project [21].

A "spin-up" phase was first performed for each site, by cycling over the available forcing years for a total of 340 years. This enables all carbon pools to stabilize and the net ecosystem production to oscillate around zero [22]. Then, a transient phase of 40 years was run to introduce anthropogenic disturbances. Finally, simulations of soil and vegetation COS and $CO_2$ fluxes were run until 2020.

Nocturnal ecosystem COS fluxes take into account plant COS uptake and soil COS uptake and production, computed with mechanistic-based models recently implemented in ORCHIDEE (see [2, 4] for a detailed description of the vegetation and soil models and Table A3 of [4]

where the nocturnal stomatal conductances adopted in ORCHIDEE tend to lower the vegetation uptake of COS at night compared to other approaches).

## Results and discussion

### A profile vs. model approach of COS nocturnal exchange rates at TRN

The diurnal and seasonal variability of the atmospheric boundary layer (ABL) depth ($z_i$) at the TRN site has been extensively investigated by [23] from lidar measurements, vertical profiles of $CO_2$ at 5, 50, 100 and 180 m along the 207 m tall tower, and $^{222}$Rn measurements at 180 m. The authors of [23] showed that, in 33.9% of cases, $z_i$ was below the top of the tower during the summer (JJA) nights of 2011 ([23], cf. Fig 5C). As shown in S1 Fig, during the months of May through August of 2019 and 2020, the highest inlet height remained within the NBL during the entire night because the nocturnal $CO_2$ monthly means at 180 m were 1.7 to 7.7 ppm (in 2019) and 2.7 to 6.2 ppm (in 2020) higher during the night (black curves, strong stratification) than during the day (blue curves, strong vertical mixing). Thus (1) the 180 m inlet did not sample air from the residual layer (RL) by night, in support of Pal et al. (2015)'s scenario B ([23], cf. Fig A1), and (2) the concentration gradients between the two levels indicate a single land use type, i.e. that they are driven by local processes.

Data collected in June-July 2020 at the TRN site (S2 Fig), although after the 2019 survey (Fig 3), should be taken as a preliminary illustration of the approach aiming at assessing monthly changes in net nocturnal exchange rates from the survey of relative vertical gradients of COS (GradCOS / GradCO$_2$). Indeed, we were unable to fully apply that approach during summer in 2020 as we did in 2019 because the IR-laser of our mini-QCL failed in August 2020. Moreover, data selection is applied to the 2020 record (S2 Fig) whereas temporal variations in the slope of linear regressions (ppt/ppm) are reported strictly on a monthly basis (Fig 3A) as when these are computed by the ORCHIDEE model (Fig 3B).

Whereas ecosystem respiration accounts for the nighttime build-up of $CO_2$ near the ground in agroecosystems during the growing season, net fluxes of COS behave in the opposite way as shown in panels A and B of S2 Fig. During the 3$^{rd}$ week of June 2020, nocturnal COS losses were also observed at 50 m and, in a lesser extent, at 100 m (data not shown). Within the nocturnal boundary layer (see $z_i$ discussion above), 180 m is the sampling height up to which the nocturnal COS drawdown almost never propagates. That is why reference air is taken at 180 m. The average nocturnal COS gradient between 5 and 180 m is generally negative whereas that of $CO_2$ is always positive (S2C Fig). When plotted against one another, the slopes of the linear regressions forced through zero exhibit an abrupt change during the second half of June 2020, from -1.17 ppt/ppm to -0.40 ppt/ppm. This finding suggests that the efficiency of the ecosystem to absorb COS from the atmosphere by night declines as the growing season proceeds. However, 2019 data shows that the latter recovers in August (Fig 3A, slope = -0.55 ppt/ppm, to be compared with the July one equal to -0.21 ppt/ppm). Nocturnal ecosystem fluxes of COS and $CO_2$ simulated by the ORCHIDEE model are strongly correlated throughout the growing season, except during a heatwave which lasted a few days in July 2019 and resulted in enhanced soil production of COS (Fig 3B). This strong correlation describes a proportionality between the $CO_2$ and COS fluxes that remains constant over all summer months, as opposite to what is found for the concentration gradients (Fig 3A) and illustrated by a different linear regression each month. If a decrease in ecosystem efficiency to absorb COS was simulated until July, followed by a recovery in August, we would not expect a single linear regression to be able to fit the simulated COS versus $CO_2$ fluxes. Hence, this simulated COS exchange, i.e., summed changes in nocturnal stomatal conductance and soil fluxes, failed to reproduce the

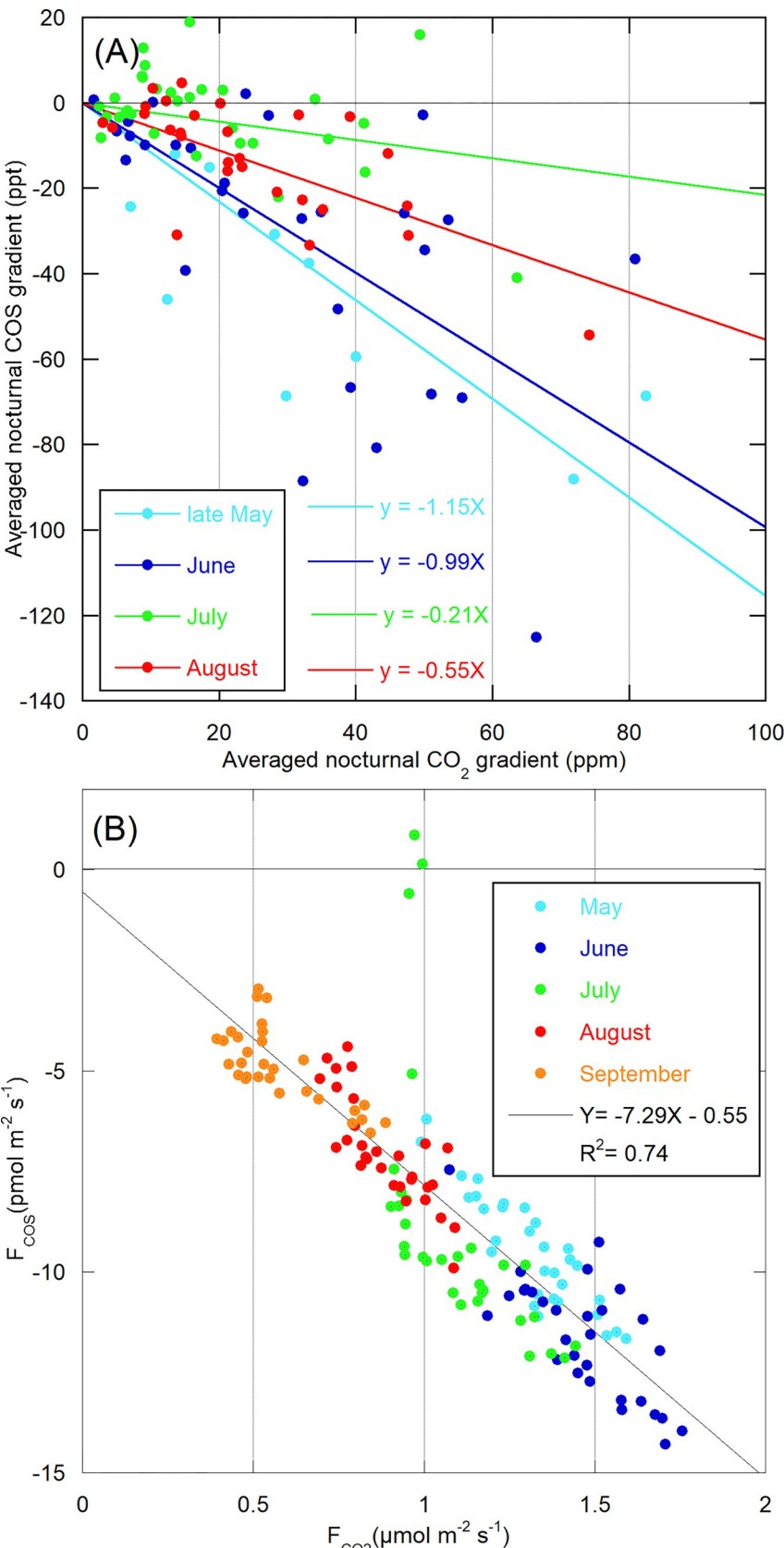

**Fig 3. Relative changes in observed COS vertical gradients and simulated fluxes at the TRN site (2019).** The nocturnal observed vertical gradients between 5 and 180m (A; $\Delta COS_{5m-180m}$) and simulated fluxes of COS (B) are plotted respectively against those of $CO_2$. Months are color coded.

decrease in the net sink of COS at TRN during the spring-summer 2019 and its recovery at the end of summer.

## Multi-year variations of COS mixing ratio and exchange rates

The GIF time series presently includes 45,000 hourly measurements of atmospheric COS mixing ratio (S3 Fig). It exhibits seasonal variations and a long-term decreasing trend as elsewhere in the northern hemisphere (e.g., https://gml.noaa.gov/dv/iadv/graph.php?code=MLO&program=hats&type=ts, [24]). Indeed, during spring both in 2020 and 2021, COS rarely exceeded 500 ppt at the GIF station, a feature corroborated by data collected at 5 m agl at the TRN site where the mini-QCL remained operative during the French COVID-19 lockdown (S4 Fig).

A plot of the updated time series from [14] of nocturnal fluxes computed with RTM at GIF site is provided in S5 Fig. After being updated, the general features put forward in [14] remain essentially the same. In 365 out of 400 cases, the site is a net sink with a median value of -5.8 pmol m$^{-2}$ s$^{-1}$. The median emission of the 35 nocturnal events recorded during about 6.5 years is 9.8 pmol m$^{-2}$ s$^{-1}$. More precisely, emission episodes were systematically recorded during the months of May through July over six consecutive years. Such emission events represent about three quarters of all emission episodes. Model simulations have been used to assess their origin.

## A RTM vs. model approach of COS nocturnal exchange rates

At GIF site, the temporal variations in nocturnal ecosystem exchange rates of COS and $CO_2$ simulated by ORCHIDEE have opposite signs (Fig 4A).

The difference between RTM fluxes and simulations is quantified in Fig 4B. Here, as we aim at finding an easy-to-use empirical function to describe this difference, May to July data shown in Fig 4C has been fit with a polynomial function of the following form applied from May 1$^{st}$,

$$Y = \frac{A * \alpha^2}{\alpha^2 + (x - x_0)^2}$$

with coefficients $x_0$, $A$ and $\alpha$ equal to 63.1 ± 0.6, 22.7 ± 0.6 and 15.3 ± 0.4, respectively. This implies that simulated ecosystem exchanges of COS at the GIF site are unable to account for observations. Furthermore, it reveals the existence of a missing source, the intensity of which culminates in late June early July. Its origin is elucidated from investigations of agricultural crops as shown below.

## A comparative study of COS exchange by wheat and rapeseed

The direction and magnitude of daytime COS exchange by wheat and rapeseed fields have been assessed in 2020 and 2021 from downwind-upwind differences in COS concentration measured at the top of the canopy ($\Delta COS_{downwind-upwind}$, S6 Fig). A first attempt carried out in 2019 served as a test of the methodological approach and indicated that rapeseed was a potential source of COS (S6 Fig). In 2021, the rapeseed and wheat fields shifted from net uptake to net emission at DOY 155 and DOY 180, respectively (S6 Fig). The significance of the

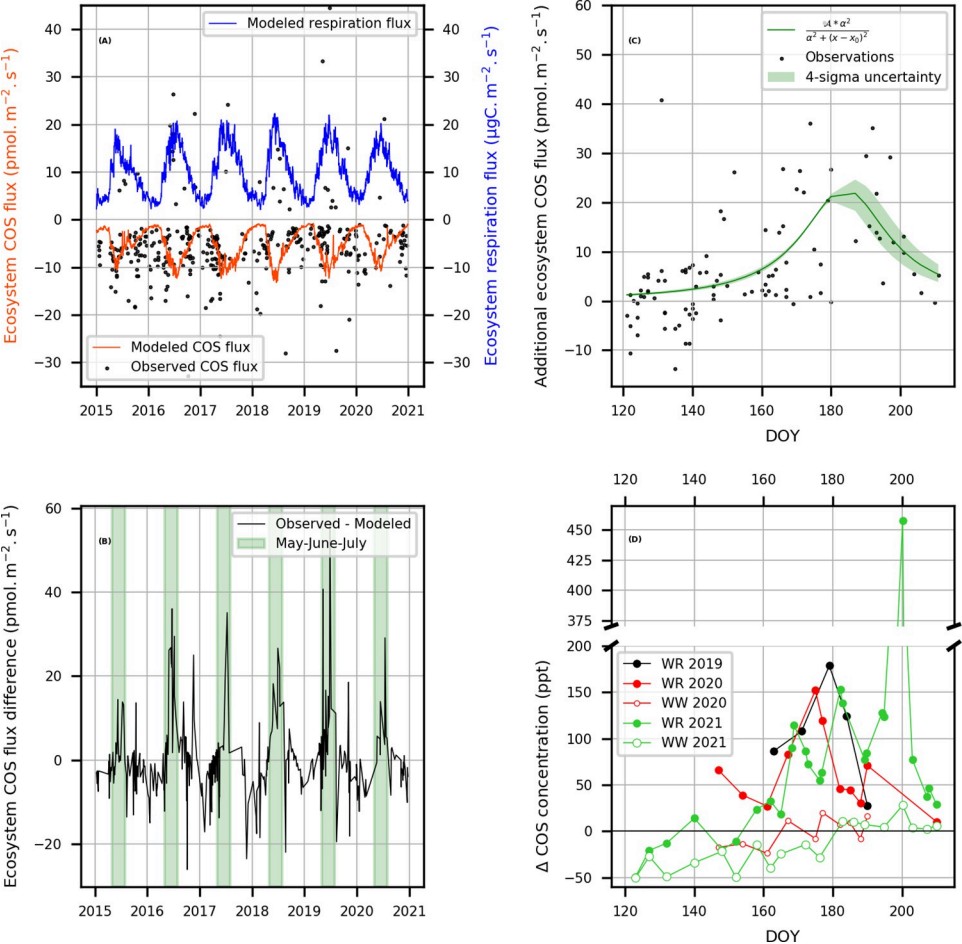

**Fig 4. Comparison between observed and simulated nighttime fluxes at the GIF site.** (A) Comparison of multi-year variations in simulated nocturnal ecosystem exchange rates of COS (red line) and $CO_2$ (blue line) with COS fluxes estimated by RTM (black dots, 2014–2020). (B) Difference between "observed" and simulated COS fluxes over 6 years. The period May to July is depicted with a green band. (C) Selected (May to July) data gathered from panel (B) then fitted with a polynomial function of the type depicted in this panel which estimates the dynamics of an additional, yet unidentified, source of COS. Fit: $x0 = 63.1 \pm 0.6$, $A = 22.7 \pm 0.6$, alpha = $15.3 \pm 0.4$, $R^2 = 0.33$. The light green shading corresponds to 4-sigma uncertainty. (D) Comparison of the dynamics of COS exchange between crops (WR: winter rapeseed; WW: winter wheat) and the atmosphere, assessed indirectly from horizontal concentration gradients (ΔCOS) downwind and upwind of selected plots (Fig 1). The full record including data collected between late March and late April are displayed in S6 Fig. In 2021, the survey of crops was interrupted before harvest, whereas in 2020, the last sample was collected after harvest. Rapeseed growth stages–year 2021: day of year (DOY) >110–140, flowering; >140–160, development of fruit; >160–190, ripening; >190–210, senescence.

differences between COS enhancements over the wheat and rapeseed fields was assessed by running a Wilcoxon rank sign test for paired data. After DOY 180, when both crops became net sources of COS, the enhancements downwind the rapeseed and wheat plots were equal to $132 \pm 125$ ppt (1 SD) and $9.1 \pm 8.2$ ppt, respectively ($P = 0.016$). In 2020, when the wheat plot shifted from net sink to net source by DOY 165, the rapeseed plot was already producing COS for at least 15 days. After DOY 165 of 2020, when both crops were sources of COS of contrasted importance, the enhancements over the rapeseed and wheat plots were equal to $78 \pm 44$ ppt and $5.4 \pm 11.2$ ppt, respectively ($P = 0.016$). Moreover, the lowest COS enhancement of the 2020 survey of the rapeseed field was measured after harvest (i.e., 10 ppt vs. 25–150 ppt during the ripening and senescence). These observations are of crucial importance

because they very likely tell that the COS source is in the plant not in the soil. Hence, it appears that the net production of COS by the rapeseed field lasted 15–25 day longer and its intensity was about 14 times higher than that of the wheat plot. The shift of wheat from net uptake to net production occurred about 2 weeks later in 2021 than in 2020, probably because crops experienced a summer drought in 2020 and wetter conditions in 2021 as shown by the soil wetness index for the Ile-de-France region available in French at https://donneespubliques. meteofrance.fr/?fond=produit&id_produit=129&id_rubrique=29 (S7 Fig). For rapeseed crops, it appears that COS production in 2020 was delayed by at least one week. During the early growth stages (DOY ≤ 140), when both plants had leaves, horizontal gradients were negative indicating that COS was taken up from the atmosphere (S6 Fig). However, the difference between medians (wheat = -28.6 ppt; rapeseed = -12.9 ppt) suggests that rapeseed plots are smaller net sinks of COS than wheat plots (P = 0.027). From the comparison of Fig 4C and 4D, it appears that COS production from rapeseed crops in the ripening and senescence phases of growth is the biogenic process that is yet unaccounted for by the vegetation and soil models implemented in the ORCHIDEE land surface model, because the temporal variations of the additional ecosystem COS flux (Fig 4C) and emissions from rapeseed plots (Fig 4D) are synchronous at the GIF site.

Maseyk et al. [8] were the first to report that wheat plots in Oklahoma in May-June 2012 act as a sink of COS during the early stages of growth (green plants), then shifted to COS release during the ripening and senescence phases. However, we are unaware of any comparison between simulated and observed fluxes in north-central Oklahoma, where Maseyk et al. [8] documented ecosystem and soil fluxes of COS. Hence, it is not known whether discrepancies exist between simulated and observed fluxes that could arise from poorly calibrated plant physiological or soil parameters rather than from COS emissions from senescent plants yet unaccounted for by models. The difference between net ecosystem COS fluxes measured by eddy covariance (EC) and soil fluxes calculated at below-canopy soil temperature and water content has been tentatively estimated, from a visual inspection of their Fig 1C, to be in the range 10–15 pmol $m^{-2}$ $s^{-1}$ during the whole senescence phase (DOY 135–145). Moreover, the authors did not report on the existence of diurnal variations in COS emission from wheat plants during the senescence phase (i.e., before harvest). Our field observations support qualitatively their results. However, here we provide evidence that the contribution at plot scale of rapeseed releasing COS during the ripening and senescence phases is much more important than that of wheat because rapeseed COS emission is stronger and lasts longer.

Evidence of COS production by rapeseed has been provided in the past from greenhouse experiments where plants in the early stage of growth (stem elongation and flowering) were exposed to unrealistic low levels of atmospheric COS (i.e., < 150 ppt, [25]) or to fungal infections [7]. Here COS production has been documented in the field from late March to late July, along each stage of growth during which farmers did not have to apply specific biochemical treatments against fungal infections. That is, our surveys of agricultural crops were carried out in conditions not perturbed by fungal infections.

A previous comparative study of biogenic volatile organic compounds (VOC) fluxes by wheat and rapeseed plants, with surveys near Paris, although at slightly different growth stages than ours (ripening and fruit development), showed that the total net VOC flux of rapeseed was about 6 times higher than that of wheat [26]. Moreover, VOC emissions of winter wheat increased twofold during the senescence stage compared to the maturation stage [27]. This is qualitatively consistent with Maseyk et al. (2014)'s observations over wheat fields [8] and ours over rapeseed crops. Although those datasets suggest rapeseed to be a stronger emitter of volatile compounds than wheat, comparison with Maseyk et al. [8] suggest that rapeseed and wheat emit COS during the senescence phase at about the same rates. However, both

approaches have estimated COS emission rates by senescent plants indirectly, i.e., from the difference between EC and soil flux measurements [8] and from the difference between fluxes estimated by the RTM method and simulated using the ORCHIDEE model (this work). There is a lack of comparative studies of COS fluxes by wheat, rapeseed and potentially by other plants using dynamic chambers as for VOC fluxes [26]. Maize would deserve special attention because it has been suggested that emission of COS from a largely senesced maize field in Bondville (USA) can disguise any leaf uptake of this gas at the end of the season [28]. The COS emissions at Bondville can account for 25% or so of the total flux (Mary Whelan, pers. com. dated October 2022).

## Implications for the global budget of COS

The COS production path in plants is not completely known. COS is thought to be produced from isothiocyanates and thiocyanate ions (SCN-) which are secondary products of the action of the myrosinase enzyme onto a large variety of forms of glucosinolates, which themselves are naturally biosynthesized in plants of the "cabbage order", Brassicales [29]. These authors also provided a list of about sixty Brassicaceae sharing the potential to form isothiocyanates. Hence, rapeseed may be only one among many Brassicaceae sharing the potential to form COS during the late stages of growth. COS is also produced from wheat [8] although it lacks the capacity to produce glucosinolates and myrosinase [30].

The empirical production function from the comparison of observed and modelled COS non-photosynthetic fluxes (Fig 4C) is suitable for inventorying COS emissions from rapeseed crops at different spatial scales. However, because its generality has not been tested yet, it should be used with caution. Indeed, we don't know if the temporal features described by this equation apply to other rapeseed fields. The magnitude of the peak emission, the date it occurs, and how quickly it wanes may vary from site to site. Nevertheless, rapeseed is mainly grown in Canada, China, the European Union, India and Australia. The total harvested areas represented about 32 million ha in 2015 [31]. Assuming that COS emissions remain invariant all day long, our empirical function, integrated over 3 months, yields a yearly total emission of $0.84 \pm 0.13$ GgS yr$^{-1}$ which, as such, does not make a significant contribution to the global budget of COS because the net uptake of COS by soils and vegetation is estimated by ORCHIDEE to be 606 GgS yr$^{-1}$ [2]. Wheat total harvested areas represent about 215 million ha (https://wheat.org/wheat-in-the-wo4rld/). Assuming that rapeseed and wheat share the same capacity of COS production during their late stages of growth, a yearly total emission of $4.7 \pm 0.7$ GgS yr$^{-1}$ is obtained. If all C3 crops would share that capacity, a weak assumption, a yearly total emission of $33.3 \pm 5.1$ GgS yr$^{-1}$ is obtained.

## Conclusions

Nocturnal enhancements of atmospheric COS observed each year during springtime and early summer in the GIF area originate from agricultural crops of wheat and rapeseed which shift from uptake to release during ripening and senescence. At the plot scale, COS concentration enhancements from rapeseed largely surpass those of wheat. At the ecosystem scale, COS emissions from agricultural crops of wheat and rapeseed either partly compensate the net nocturnal uptake of COS by vegetation and soil, as observed indirectly from measurements of COS and $CO_2$ vertical gradients at the TRN site, or largely surpass it as the RTM and model joint approach shows. A COS empirical production function is proposed which generality needs to be tested in other rapeseed fields especially in terms of duration and magnitude of the peak emission. The role of other plants of the "cabbage order" needs also to be addressed experimentally in the fields with dynamic chambers or by eddy covariance. Although our field

observations support qualitatively those of [8], some inconsistencies remain as to the relative importance of COS emissions from senescent wheat at the plot scale on both sides of the Atlantic. For now, it is suggested that emissions from rapeseed alone and C3 crops in general cannot account for the missing source of COS in the global budget of this gas.

## Supporting information

**S1 Fig. $CO_2$ vertical gradient, per month and per daily period at the TRN site for the years 2019 (upper plot) and 2020 (lower plot).** These are standard ICOS products generated by the ICOS database.
(PDF)

**S2 Fig. Variations in mixing ratios and in relative COS nocturnal vertical gradient.** (A) $CO_2$ and (B) COS (OCS) mixing ratios measured at 5 and 180 m roughly on an hourly basis at the TRN site in June-July 2020. (C) Correlation of COS (OCS) and $CO_2$ averaged nocturnal vertical gradients measured on a daily basis. Those linear regressions forced through zero are not calculated strictly on a monthly basis but after data selection because the transition from one regime (slope = -1.17 ppt/ppm) to another (slope = -0.4 ppt/ppm) took place the night of June 21$^{st}$ to 22$^{nd}$ as shown in panels A and B.
(PDF)

**S3 Fig. Multi-year variations in COS mixing ratio at GIF with hourly resolution.** The data gaps in summer/early autumn of 2017 and spring 2020 being a failure of the Entech preconcentrator and the consequence of the French lockdown, respectively. The full COS records are now available from https://doi.org/10.14768/6800b065-dcec-4006-ada5-b5f62a4bb832.
(PDF)

**S4 Fig. Hourly variations in COS mixing ratio at the GIF (7m agl) and TRN (5m agl) sites.** The mini-QCL remained operative at TRN during the French lockdown while GC measurements at GIF were stopped for about two months.
(PDF)

**S5 Fig. Multi-year variations in COS exchange rates at the GIF site.** These are nocturnal COS (OCS) fluxes obtained by the Radon Tracer Method.
(PDF)

**S6 Fig. Variations during the growing season in uptake/emission regimes by wheat and rapeseed assessed from horizontal gradients of COS.** The difference in COS concentrations measured downwind and upwind of selected plots is plotted against day of year (DOY). Measurements were carried out in the morning between 9:00 and 11:00 (local time), during no rainy days and roughly in similar meteorological conditions according to wind speed ($2 < $ WS $ < 14$ km h$^{-1}$, mean = 6 km h$^{-1}$, SD = 3.6 km h$^{-1}$). In 2021, the survey of crops was interrupted before harvest, whereas in 2020, the last sample was collected after harvest. The lag in 2021 of the shift from net uptake to net production (upt-to-prod) for either WW or WR is depicted by an horizontal double arrow. Rapeseed growth stages—year 2021: DOY<110, inflorescence emergence and elongation; >110–140, flowering; >140–160, development of fruit; >160–190, ripening; >190–210, senescence. We have zoomed in the May-to-July period in Fig 4D.
(PDF)

**S7 Fig. Soil wetness index (SWI) for the Ile-de-France region.** The SWI is a soil moisture index documented in the scientific literature. It represents, over a depth of about two meters, the state of the water reserve of the soil in relation to the useful reserve (water available for

plant nutrition). Plots downloaded from https://donneespubliques.meteofrance.fr/?fond=produit&id_produit=129&id_rubrique=29. First, we selected Bulletin climatique mensuel régional (à partir de janvier 2020), then Ile-de-France region from the drop-down menu, then we downloaded reports for the months of July 2020 and July 2021, then compared graphs entitled "Indice d'humidité des sols" in page 4 of 5). Upper panel: March $1^{st}$ to July $31^{st}$, 2020. Lower panel: March $1^{st}$ to July $31^{st}$, 2021. Refer only to the purple curves.
(PDF)

## Acknowledgments

SB expresses its special thanks to Mark Zahniser at Aerodyne Research for its unconditional and enthusiastic support during operation of the mini-QCL. We thank Nicolas Vuichard and Tanguy Martinez for preparing the ERA5 forcing files for the ORCHIDEE model and the ICOS data products displayed in the supplements, respectively. We wish to thank the two reviewers, including Mary Whelan, for their helpful suggestions to improve the paper.

## Author Contributions

**Conceptualization:** Sauveur Belviso, Camille Abadie, David Montagne, Fabienne Maignan.

**Data curation:** Sauveur Belviso, Camille Abadie.

**Funding acquisition:** Michel Ramonet, Philippe Ciais.

**Investigation:** Sauveur Belviso, Camille Abadie, David Montagne, Dalila Hadjar, Didier Tropée, Laurence Vialettes, Victor Kazan, Marc Delmotte, Fabienne Maignan, Marine Remaud, Michel Ramonet, Morgan Lopez.

**Methodology:** Sauveur Belviso, Camille Abadie, David Montagne, Marc Delmotte, Fabienne Maignan, Michel Ramonet, Morgan Lopez.

**Software:** Fabienne Maignan, Marine Remaud, Camille Yver-Kwok.

**Writing – original draft:** Sauveur Belviso, Camille Abadie, David Montagne.

**Writing – review & editing:** Marc Delmotte, Fabienne Maignan, Marine Remaud, Michel Ramonet, Morgan Lopez, Camille Yver-Kwok, Philippe Ciais.

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
