## [Decision Letter · Decision Letter 0]

17 Aug 2022

PONE-D-22-03159Carbonyl sulfide (COS) emissions in two agroecosystems in central FrancePLOS ONE

Dear Dr. Belviso,

Thank you for submitting your manuscript to PLOS ONE. After careful consideration, we feel that it has merit but does not fully meet PLOS ONE’s publication criteria as it currently stands. Therefore, we invite you to submit a revised version of the manuscript that addresses the points raised during the review process.

We look forward to receiving your revised manuscript.

Kind regards,

Tanvir Shahzad

Academic Editor

PLOS ONE

Journal Requirements:

2. We note that Figure 1 in your submission contain [map/satellite] images which may be copyrighted. All PLOS content is published under the Creative Commons Attribution License (CC BY 4.0), which means that the manuscript, images, and Supporting Information files will be freely available online, and any third party is permitted to access, download, copy, distribute, and use these materials in any way, even commercially, with proper attribution. For these reasons, we cannot publish previously copyrighted maps or satellite images created using proprietary data, such as Google software (Google Maps, Street View, and Earth). For more information, see our copyright guidelines: http://journals.plos.org/plosone/s/licenses-and-copyright.

Reviewers' comments:

Reviewer's Responses to Questions

**Comments to the Author**

1. Is the manuscript technically sound, and do the data support the conclusions?

Reviewer #1: Partly

2. Has the statistical analysis been performed appropriately and rigorously? 

Reviewer #1: No

3. Have the authors made all data underlying the findings in their manuscript fully available?

Reviewer #1: Yes

4. Is the manuscript presented in an intelligible fashion and written in standard English?

Reviewer #1: Yes

5. Review Comments to the Author

Reviewer #1: The study by Belviso et al. aims to examine the source and sink dynamics of carbonyl sulfide over wheat and rapeseed fields in the Paris region. The authors used two independent methods to infer nighttime COS fluxes (the radon tracer method and the gradient method) and the horizontal gradient method to qualitatively assess daytime COS fluxes. Although these methods each have their own gaps and the authors did not provide an ecosystem budget, daytime COS enhancements at downwind locations seem to provide evidence that the rapeseed field was a strong COS source during ripening and senescence. In my opinion, this is the main novelty and probably the most solid part of the work. However, the interpretation of the results suffers from several conceptual confusions and methodological weaknesses. There are three major issues that need to be addressed and some room for improvement in other aspects.

First, the authors attribute COS emissions to rapeseed plants throughout the discussion. While the observed COS enhancement downwind of the rapeseed field (Fig. 5D) indicates the presence of a COS source in this ecosystem, it cannot tell whether this source is in the plants or the soil. As the authors are aware, both rapeseed plants (Bloem et al., 2012) and agricultural soils (Maseyk et al., 2014; Whelan et al., 2015) have been found capable of emitting COS. Without in situ soil flux or leaf flux measurements, there is no reason to exclude the possibility of a COS source in soils.

Second, the authors interpret the difference between fluxes simulated by the ORCHIDEE model and derived from COS concentration observations simplistically as “additional ecosystem COS flux” (Fig. 5B). They then interpreted the exceptionally high difference between observed and modelled fluxes in May–July as evidence for a missing COS source (Fig. 5C). This is fraught with issues because many factors lead to model–data discrepancies. For example, the model’s default set of plant physiological parameters determining COS fluxes may differ from the parameters measured in the field. This is because land surface models typically represent plants at the functional type level, not at the species level. When models do represent major crop species, they may use a general set of parameters and not have the granularity to finely resolve crop varieties actually grown in the field, unless they are customised to do so. Because simulated nighttime COS fluxes critically depend on nighttime stomatal conductance (Kooijmans et al., 2017), model–data discrepancies could also be attributed to misrepresented nighttime stomatal conductance (which we know to be the case for many models, e.g., Kooijmans et al., 2021). In addition, the large discrepancy in May–July could be due to misrepresented soil moisture limitation on stomatal conductance. There is insufficient evidence to pin this discrepancy on a missing COS source.

Third, given the patchy nature of the land use map (Fig. 1), it is unclear how the authors achieved a clear separation of wheat vs. rapeseed influences on the downwind COS enhancements. Spatial heterogeneity presents a bigger problem for the TRN site. Because measurements at 180 m cover a greater footprint area than measurements at 5 m, the fluxes calculated from concentration gradients between the two levels are not specific to a single land use type.

Given these weaknesses, it seems more fruitful for the manuscript to focus on results of relatively high confidence, such as large COS emissions from the rapeseed field. If the authors want to make the claim of a large COS source in rapeseed plants, they need to measure soil or leaf COS fluxes, not relying on atmospheric COS concentrations to resolve what cannot be resolved from a top-down point of view. They would also need to calibrate parameters of the ORCHIDEE model against those measured on wheat and rapeseed plants in the field, before suggesting a missing COS source (but this would not be needed if a COS source is directly observed in the field). Alternatively, the interpretation must be revised to reflect alternative explanations.

The presentation of the manuscript could be streamlined to benefit readers who may not be well-versed in carbonyl sulfide research or flux quantification methods. Rather than confronting the reader with the current uncertainty in COS budgets, the introduction could start with a brief statement on why we care about COS and why there is a need to examine its sources and sinks in croplands. It would be better to list the description of the three flux quantification methods—horizontal gradient method for daytime fluxes at GIF, radon tracer method for nighttime fluxes at GIF, and vertical gradient method at TRN—under separate headings to avoid potential confusion.

Lastly, I commend the authors for making the data openly available with no strings attached. Please consider putting the data set in long-term archives (for example, Zenodo) so that it can be easily searched, accessed, and cited.

Specific comments

L23: “exchange fluxes” -> “fluxes”

L23–24: “simulated by vegetation and soil models, implemented in the ORCHIDEE land surface model” -> “simulated by the ORCHIDEE land surface model”

L32: Readers would not have known what the “profile vs. model” approach means at this point. Explain the approach and to which modelled variables the profile is compared.

L34: “were used for that same purpose” - Unclear from the context. Specify the purpose.

L39–40: “the nocturnal net COS uptake gradually weakens” - The weakening of nocturnal net COS uptake does not imply the existence of a source. It could be due to the reduction of nocturnal stomatal conductance in response to increased atmospheric and soil water stress in the late growing season.

L43–44: The downwind enhancement of COS concentration over the wheat field is too small to indicate a robust COS source. How does the difference compare to uncertainty from measurement and atmospheric transport?

L53–54: “yet estimates differ by as much as a factor of 5.6 as the recent review of [1] shows” - If estimates based on NPP scaling (which we now know is incorrect understanding) are excluded, the rest of the estimates based on GPP scaling or mechanistic models vary by a factor of two only.

L66–67: “vascular plants could play also a role in COS production” - I would add that this happens for a limited number of species (Geng et al., 2006), during senescence (Maseyk et al., 2014), or with fungal infections (e.g., Bloem et al., 2012), lest readers think it is a common behaviour.

L103–107: This is a convoluted sentence. Simplify it.

L105–107: “the fields over which the order of magnitude of COS fluxes were assessed indirectly from downwind minus upwind horizontal concentration gradients along the wind direction” - This seems to belong to the measurement methods, not site description.

L107–119: It would be better to separate the description of crop planting and harvesting from that of the monitoring station.

L116–117: Was the rapeseed field fertilised with sulfur as well?

L128: “They were documented in the morning between 9:00 and 11:00 (local time)” - How would you ensure that the concentration gradient was not affected by the residual nocturnal boundary layer?

L129: “from flask-air samples collected in pairs each upwind and downwind” - Did you measure the wind speed and direction? Was there any change in the wind regime during sample collection? Did the downwind sampling location receive influences from places other than the upwind sampling location (i.e., how were the back-trajectories like?)

L174–175: This method implicitly assumes that COS and CO2 have the same aerodynamic conductance. This assumption may not hold because we know that COS and CO2 have different molecular diffusivities in the air. Please consult Rastogi et al. (2018) J. Geophys. Res. Biogeosci. for the calculation of aerodynamic conductance and a rigorous application of the gradient method. You may also want to consider the storage fluxes depending on how fast the concentration gradients change in time (see Eq. 2 in Rastogi et al., 2018).

L226–227: “the site is a net sink” - Only at night. The radon tracer method does tell anything about daytime fluxes.

L229–230: “More precisely, emission episodes were systematically recorded during the months of May through July over six consecutive years.” - What drives nighttime emissions? Did these episodes correlate with high temperatures or high VPD?

L234–243: This paragraph seems to belong to the methods.

L256–257: “This suggests that the efficiency of the ecosystem to take up COS from the atmosphere by night significantly lowers as the growing season proceeds.” - As stated earlier, this change could result from a reduction in nocturnal stomatal conductance and does not necessarily indicate the presence of a COS source.

L262–263: “a heatwave which lasted only for few days in July 2019 and resulted in enhanced soil production of COS” - Is there any observational evidence to indicate that COS is produced from the soil, not the vegetation?

L294: As stated earlier, the discrepancy between simulated and observed fluxes is not enough evidence to suggest the presence of unaccounted COS emissions, because it could arise from poorly calibrated plant physiological or soil parameters.

L302: At the ecosystem scale, “in situ” usually refers to eddy covariance flux measurements. Suggest removal.

L313: Are the numbers after the plus/minus sign one standard deviation?

L317: “5.4 ± 11.2 ppt” - It does not seem that the COS enhancement over the wheat field is statistically significant. I suggest running a statistical test for the significance of the difference (e.g., a t-test for paired samples).

L318–320: “Hence, it appears that the net production of COS by the rapeseed field lasted longer (15-25 day longer) and its intensity is about 14 times higher than that of the wheat plot.” - The downwind–upwind concentration enhancement depends on both COS emission rates and wind speed and direction. Without running an inversion or a dispersion model to infer the fluxes, concentration difference alone is not a good indicator of fluxes.

L320–322: “The shift of wheat from net uptake to net production took place about 2 weeks later in 2021 than in 2020, likely because crops experienced a summer drought in 2020 and wetter conditions in 2021” - This seems an important point buried in the middle of the text. Could you test the influences of heatwaves or drought on emission episodes?

L322: “soil humidity index” - This needs a definition. Or, could you convert it to volumetric soil water content?

L358–371: These sentences on VOC emissions do not seem relevant. We do not know if COS production is linked to the production of sulfur-free VOCs at the molecular and cellular levels. Suggest removal.

L376–399: This paragraph feels disorganised. I would separate it into one paragraph on the biochemical mechanisms of COS production and another on the upscaled potential global budget of COS emissions from wheat and rapeseed fields.

L404: “COS emissions” -> “COS concentration enhancements”, to be precise.

L407: “net non-photosynthetic uptake” -> “net nocturnal uptake”. Daytime soil fluxes are also part of “non-photosynthetic” fluxes, but we do not know their magnitude since they are not measured.

Fig. 1: Please indicate downwind and upwind sampling locations.

Fig. 3: This plot of concentration time series does not directly inform the main points about fluxes. Suggest offloading it to the Supplement.

Fig. 4: Report CO2 fluxes in molar units because COS fluxes are in pmol m^-2 s^-1.

Fig. 5: This is a comparison between observed and simulated nighttime fluxes. They should not be interpreted as the ecosystem budget, unless you have data to show that daytime emissions behave the same as nighttime emissions.

6. PLOS authors have the option to publish the peer review history of their article (what does this mean?). If published, this will include your full peer review and any attached files.

Reviewer #1: No

---

## [Author Response · Author response to Decision Letter 0]

13 Sep 2022

please refer to to file named "response to reviewers"

---

## [Decision Letter · Decision Letter 1]

6 Nov 2022

PONE-D-22-03159R1Carbonyl sulfide (COS) emissions in two agroecosystems in central FrancePLOS ONE

Dear Dr. Sauveur Belviso,

Thank you for submitting your manuscript to PLOS ONE. After careful consideration, we feel that it has merit but does not fully meet PLOS ONE’s publication criteria as it currently stands. Therefore, we invite you to submit a revised version of the manuscript that addresses the points raised during the review process.

We look forward to receiving your revised manuscript.

Kind regards,

Tanvir Shahzad

Academic Editor

PLOS ONE

Journal Requirements:

Reviewers' comments:

Reviewer's Responses to Questions

**Comments to the Author**

1. If the authors have adequately addressed your comments raised in a previous round of review and you feel that this manuscript is now acceptable for publication, you may indicate that here to bypass the “Comments to the Author” section, enter your conflict of interest statement in the “Confidential to Editor” section, and submit your "Accept" recommendation.

Reviewer #1: (No Response)

Reviewer #2: (No Response)

2. Is the manuscript technically sound, and do the data support the conclusions?

Reviewer #1: Yes

Reviewer #2: Yes

3. Has the statistical analysis been performed appropriately and rigorously? 

Reviewer #1: Yes

Reviewer #2: Yes

4. Have the authors made all data underlying the findings in their manuscript fully available?

Reviewer #1: Yes

Reviewer #2: Yes

5. Is the manuscript presented in an intelligible fashion and written in standard English?

Reviewer #1: Yes

Reviewer #2: Yes

6. Review Comments to the Author

Reviewer #1: The revised manuscript has been greatly improved in clarity and soundness. The description of methods is much easier to follow now. The interpretation of results is put on more solid ground. I only have a few minor comments.

L114–116: “A proxy of the ratio of nocturnal ecosystem fluxes (FCOS/FCO2) was calculated from nighttime vertical gradients of COS and CO2 (GradCOS/GradCO2 from data collected at 5 m and 180 m height)” - It may help to give the readers a reason why you calculate the gradient between 5 m and 180 m instead of that between 5 m and 50 m.

L117: “CO2 flux-gradients” -> “CO2 fluxes and gradients”

L237: “The authors of ref. [23] showed that ...”

L301: “phase” is ambiguous here. Saying that they have opposite signs should suffice.

Fig. 3A: Use English decimal points in the legend.

Fig. 4C: The data points don’t appear to follow the prescribed equation very well. Perhaps a non-parametric fit such as local regression would represent the temporal trend of “additional fluxes” more faithfully.

L430–432: "The empirical production function from the comparison of observed and modelled COS non-photosynthetic fluxes (Fig 5C) is suitable for inventorying COS emissions from rapeseed crops at different spatial scales.” - The generality of this “empirical production function” needs to be tested. We don’t know if the temporal features described by this equation apply to other rapeseed fields. The magnitude of the peak emission, the date it occurs, and how quickly it wanes may vary from site to site.

Reviewer #2: Dear Dr. Belviso, et al.,

OCS fluxes over agricultural fields are an important conundrum in global OCS modeling. Some crops (e.g. rapeseed, investigated here) emit OCS, and some soils emit OCS when hot and dry. A multi-year investigation using multiple methods furthers our understanding of the complexity of crops and OCS. This project is well-conceived and my comments are therefore minor.

Abstract: It might bear mentioning that the GIF site has observations going back to 2014. This would be helpful to alert the reader to the excellent dataset, the length of record in some of the figures, and the multi-year usefulness of this site.

Data: While I appreciated being able to download the excel spreadsheet of observational data quickly, the excel file itself has no metadata and I’m not sure about the longevity of mycore.core-cloud.net. The data at data.ipsl.fr/respository is obviously preferable, though these files don’t appear to have metadata either. One could say, “the data is described in the paper” which is true. What would be good is to put the citation in the header of the file and maybe (if you are feeling generous) include a brief description of the variables.

Supplemental: There is no limit to the length of articles for this publication. I’m not sure why we would therefore have a supplemental. Why not have all the figures in one, crisp narrative? The supplemental figures are well-rendered.

(Very) minor comments

25: Not yet accounted for.

87-88: Is this a statement that needs to be made in the published work? This almost feels like it should go at the end under acknowledgements, but I don’t know about the shifting requirements of environmental research where you are.

102: Original image of what?

107: Is this an archived website that will be there in 20 years? Otherwise, include a brief description.

115-116: Please write the equation to calculate gradients. Was it really concentration at 5 m – concentration at 180 m? Why collect at 4 heights? When people talk about the flux gradient method, normally we think about something a little more complicated, e.g. see section 2.6 of Griffis et al., 2005, https://doi.org/10.1016/j.agrformet.2005.10.002.

117-121: [21; 4[ (UTC)? This sentence needs to be revised. There are too many ideas in it. This makes it sound like TRN was only used for nocturnal gradients.

190, 192: You could put in the latin names for plant scientists of the future.

197-198, 254: Is this a statement required of the funding source?

200: It might be cultivars, not cultures.

232: Quotes are unnecessary /confusing here.

247: How do you get a single land use type from a pair of concentrations?

278-281: There are logical leaps to get to this statement. They just need to be spelled out.

298: It might be good to explain what you think the emissions episodes are caused by here.

311: The emission is unidentified, but this is the results and discussion section. There are plenty of candidates in the literature you could mention. Why do this? It will inspire other folks to try and figure it out and they can start with a hypothesis to test.

352-354: This conclusion needs to be given greater prominence.

366-367: Is the difference in uptake related to a difference in GPP?

379-382: I thought you just demonstrated that there’s a missing plant-based process not accounted for by models?

385: Kadmiel Maseyk can send you the data so you do not have to read it off of Fig 1.

374-419: This paragraph is a beast. It would be easier to understand what is going on by breaking it up into more digestible pieces.

400: Crops seem by their nature to be perturbed?

418-419: The OCS emissions at Bondville can account for 25% or so of the total flux.

410: I would do a find-and-replace for OCS to make sure it’s all COS (or all OCS).

428-429: It seems like this paragraph should start, “The COS production path in plants is not completely known.” The way it starts now, it seems as though the production path has extensive research, but then we end by saying we have no idea what’s going on with wheat.

446-459: This conclusion is okay, but is anemic compared to the colossal effort and an excellent dataset. Where do you want it to go now?

Thanks for moving this idea forward,

Mary Whelan

7. PLOS authors have the option to publish the peer review history of their article (what does this mean?). If published, this will include your full peer review and any attached files.

Reviewer #1: No

Reviewer #2: **Yes: **Mary Whelan

---

## [Editor Report · Decision Letter 2]

21 Nov 2022

Carbonyl sulfide (COS) emissions in two agroecosystems in central France

PONE-D-22-03159R2

Dear Dr. Sauveur Belviso,

We’re pleased to inform you that your manuscript has been judged scientifically suitable for publication and will be formally accepted for publication once it meets all outstanding technical requirements.

Kind regards,

Tanvir Shahzad

Academic Editor

PLOS ONE
---

## [Editor Report · Acceptance letter]

23 Nov 2022

PONE-D-22-03159R2 

Carbonyl sulfide (COS) emissions in two agroecosystems in central France 

Dear Dr. Belviso:

I'm pleased to inform you that your manuscript has been deemed suitable for publication in PLOS ONE. Congratulations! Your manuscript is now with our production department. 

Kind regards, 

on behalf of

Dr. Tanvir Shahzad 

Academic Editor

PLOS ONE